# *Opuntia dillenii* Haw. Polysaccharide Promotes Cholesterol Efflux in THP-1-Derived Foam Cells via the PPARγ-LXRα Signaling Pathway

**DOI:** 10.3390/molecules27248639

**Published:** 2022-12-07

**Authors:** Heng Li, Zhenchi Huang, Fuhua Zeng

**Affiliations:** 1School of Food Science and Engineering, Lingnan Normal University, Zhanjiang 504048, China; 2School of Life Science and Technology, Lingnan Normal University, Zhanjiang 524048, China

**Keywords:** *Opuntia dillenii* Haw. polysaccharide, cholesterol efflux, ABCA1, PPARγ-LXRα, THP-1-derived foam cells

## Abstract

There is increasing evidence supporting a role for enhanced macrophage cholesterol efflux in ameliorating atherosclerosis. *Opuntia dillenii* Haw. polysaccharide (ODP-Ia), the most important functional component obtained from *Opuntia dillenii* Haw. stem, has anti-atherosclerosis effects. Therefore, we propose that ODP-Ia could promote cholesterol efflux via the PPARγ-LXRα signaling pathway. In this study, THP-1 foam cells derived from macrophages were treated with different concentrations of ODP-Ia, GGPP (antagonist of LXRα) and GW9662 (antagonist of PPARγ), with or without 15 nmol ODP-Ia. The total cholesterol content in the cells was measured. The mRNA of *ABCA1*, *ABCG1*, *PPARγ*, *LXRα* and their protein levels in the foam cells were detected by RT–PCR and Western blot, respectively. The results showed that ODP-Ia plays a role in significantly promoting cholesterol efflux (*p* < 0.05) by upregulating the expression of ABCA1, ABCG1, SR-BI, PPARγ, PPARα and LXRα. Meanwhile, PPARγ and LXRα antagonists dramatically interfered the cholesterol efflux mediated by ODP-Ia (*p* < 0.05) and dramatically inhibited the upregulating effect of ODP-Ia on the expression of PPARγ, LXRα, ABCA1 and ABCG1 at both protein and mRNA levels (*p* < 0.05). In conclusion, ODP-Ia promotes cholesterol efflux in the foam cells through activating the PPARγ-LXRα signaling pathway. This bioactivity suggested that ODP-Ia may be of benefit in treating atherosclerosis.

## 1. Introduction

The balance of cholesterol metabolism must be precisely modulated to maintain the homeostasis of the body in peripheral cells, especially in macrophages [1]. Macrophages can take in not only the remaining free cholesterol in endometrial cells, but also that from cell debris and lipoprotein substances, and this intake is not suppressed by the negative feedback regulation mechanism [2]. Once the intake exceeds the amount of outflow, macrophages with excess lipid will initiate pathological processes, foam or apoptosis. Monocytes in the blood circulation are transformed into macrophages to phagocytize lipids accumulated in the vascular wall, which can be regarded as a self-protection mechanism of the human body. However, when consumed in large quantities, the human body must have a strong ability to transport these lipids out in order to maintain the balance of lipid metabolism. Therefore, transporting lipids, that is, promoting cholesterol outflow, directly affects the survival and development of macrophages [3,4,5].

There are three main pathways of cholesterol efflux from peripheral cells: water-soluble diffusion, Cavenger receptor class B type 1 (SR-BI) mediation and ATP binding cassette transporter A1 (ABCA1) mediation, among which ABCA1 mediation is considered to be the most important [6,7,8]. In this pathway, ABCA1 mediates intracellular free cholesterol and phospholipids to Apolipoprotein A-I (apoA-I), and synthesizes immature lipid or lipid-free HDL precursors into newborn HDL [9]. HDL plays an important role in the process of cholesterol efflux [10]. It transports intracellular cholesterol from peripheral cells to liver, and then excretes bile through feces in vitro. Many studies have confirmed that high HDL in plasma is a favorable factor to prevent and inhibit the formation and development of atherosclerosis [11,12,13]. ApoA-I is the main apolipoprotein component of HDL, accounting for about 70% of the protein composition in mature HDL. Therefore, apoA-I plays an important role in promoting cholesterol outflow as a transporter. ABCA1 is regulated by many factors in vivo, including the Liver X receptor (LXR) family, Peroxisome proliferator-activated receptor (PPAR) family and inflammatory factor TGF-beta, which upregulate the expression of *ABCA1* gene, tumor necrosis factor-α (TNF-α), Interleukin 1 beta (IL-1β) and tumor necrosis factor-γ (TNF-γ), which downregulate the expression of ABCA1 gene [14]. The LXR family receptor is considered to be the main nuclear receptor, and regulates the expression of *ABCA1* gene because of LXR/RXR, a heterodimer composed of liver X receptors and retinoid X receptor, can recognize the DR-4 element on the promoter of *ABCA1* gene [15,16]. PPAR family receptors are also the regulators in many studies, but now it is considered that PPAR family receptors cannot directly regulate the expression of the *ABCA1* gene promoter, but rather indirectly regulate it through LXR family receptors [17]. Therefore, PPARγ-LXRα is a well-documented ABCA1 regulatory pathway [18,19].

Prickly pear *Opuntia dillenii* Haw., located in the tropical or subtropical area, is a species from the cactus family. It is a plant with high economic value, used as fodder, vegetables and fruit [20]. In addition, prickly pear *Opuntia dillenii* Haw is a medicinal plant with diverse pharmacological activities, including antioxidant, anti-tumor, anti-obesity [21], neuroprotective, hepatoprotective, hypotensive, hypolipidemic [22] and hypoglycemic [23] effects, and can also attenuate acetic acid-induced ulcerative colitis [24] and cadmium-induced liver injury in mice [25].

Preliminary work in our laboratory has confirmed that *Opuntia dillenii* Haw. polysaccharide ODP-Ia has good cholesterol-regulating effects, can significantly improve the liver and vascular lesions of hyperlipidemic rats and can reduce the degree of aortic atherosclerotic plaque lesions [26], but its specific molecular mechanism is not yet clear.

Therefore, this study, by comparing the effects of different doses of ODP-Ia on *ABCA1*, ATP binding cassette transporter G1 (*ABCG1*), *SR-BI*, *PPARγ*, *PPARα*, *LXRα* genes and protein expression in the model, investigated whether ODP-Ia played a role in promoting cholesterol efflux through the regulation of these cholesterol efflux-related genes. Meanwhile, the study explored the regulatory role of PPARγ-LXRα pathway in the process of ABCA1-mediated cholesterol efflux.

## 2. Results

### 2.1. Foam Cell Model Verification

Under normal conditions, THP-1 cells grow in suspension and are round or oval-shaped. After 24 h of PMA stimulation, the cells acquired a long spindle-like morphology with pseudopodia, grew adherently, and differentiated into macrophages (Figure 1A). After stimulation with 50 ng/mL ox-LDL for 48 h, the macrophages produced a large amount of lipids and enlarged and developed into foam cells (Figure 1B). The intracellular lipid droplets could be stained with Oil red O. Once stained, the droplets turned red and were wrapped around the nucleus like rings, indicating that THP-1 macrophage-derived foam cells were successfully established.

### 2.2. Effects of ODP-Ia and Ezetimibe on the Viability of THP-1 Macrophage-Derived Foam Cells

As shown in Figure 2A, no significant difference in relative cell viability was observed between the 5, 10 and 20 nmol/L ODP-Ia groups and the control group (without ODP-Ia). Therefore, 10 nmol/L was used as the optimal treatment concentration of ODP-Ia, and 5, 10 and 15 nmol/L were used as the low-, medium- and high-concentration doses.

In another study [27], the optimal treatment concentration of ezetimibe that had an effect on THP-1 macrophage-derived foam cells was 3 μmol/L. As shown in Figure 2B, ezetimibe had no significant inhibitory effect on cells at concentrations of <9 μmol/L (no significant difference in relative survival compared with the control group). Therefore, the optimal treatment concentration of ezetimibe was 3 μmol/L.

### 2.3. Effects of ODP-Ia on Cholesterol Outflow from Foam Cells during apoA-I-Mediated Cholesterol Efflux

As shown in Figure 3A,B, when treated with apoA-I, the total cholesterol content of the THP-1 macrophage-derived foam cells decreased in a time- and concentration-dependent manner. Compared with the control group, the reduction effect of the low dose of 5 mg/mL was already significant (*p* < 0.05). However, no significant difference was observed between the 10 and 15 mg/mL groups. To obtain a more significant effect, we selected 20 mg/mL as the optimal treatment dose. The total cholesterol content of the THP-1 macrophage-derived foam cells decreased significantly with increasing treatment time (6–36 h) after treatment with 20 mg/mL apoA-I. The maximum reduction effect was observed after 24 and 36 h of treatment, but no significant difference was observed between the 24 and 36 h groups. Therefore, 24 h was chosen as the optimal treatment time. As shown in Figure 3C, compared with the control group, the total cholesterol content of the THP-1 macrophage-derived foam cells in the apoA-I group significantly decreased (*p* < 0.05). After pretreatment of the foam cells with ODP-Ia or ezetimibe for 2 h, the effect of apoA-I-mediated cholesterol efflux became stronger, although this effect was less effective than the effect of ezetimibe in the positive control group. The aforementioned results suggested that ODP-Ia can enhance apoA-I-mediated cholesterol efflux from foam cells.

### 2.4. Effects of ODP-Ia on the Cholesterol Outflow in Foam Cells

As shown in Figure 4, the cholesterol content of the foam cells group was significantly higher than that of the blank group (*p* < 0.05), which means the THP-1 cells were foamed successfully. After ODP-Ia intervention, the foamed effect was inhibited, and the cholesterol content in the foam cells showed a downward trend. These results indicate that ODP-Ia can promote cholesterol efflux in foam cells, and that 15 nmol/L is an effective dose.

### 2.5. Effects of ODP-Ia on the mRNA Expression of ABCA1, ABCG1, SR-BI, PPARγ, PPARα and LXRα in Foam Cells

As shown in Figure 5, the expression of *PPARγ*, *PPARα*, *LXRα*, *ABCA1*, *ABCG1* and *SR-BI* mRNA in THP-1-derived foam cells treated with 50 µg/mL ox-LDL decreased significantly. ODP-Ia intervention (48 h) effectively increased the expression of these genes, and there was a significant dose effect (*p* < 0.05). However, the effect of this improvement was lower than that of the positive control group (*p* < 0.05).

### 2.6. Effects of ODP-Ia on the Protein Expression of PPARγ, PPARα, LXRα, ABCA1, ABCG1 and SR-BI in Foam Cells

The Western blot results were captured by the gel imaging system, and the density values of each bands were quantified by the Quantity One analysis software of BioRad company. As shown in Figure 6, THP-1 cells could express PPARγ, PPARα, LXRα, ABCA1, ABCG1 and SR-BI after being induced by PMA, but the expression of these proteins was obviously weakened after ox-LDL treatment. After further intervention with ODP-Ia and Ezetimibe, the expression of these proteins increased gradually, and the enhancement was dosage dependent. The results of band-density quantification showed that the effect of high-dose ODP-Ia intervention was significantly higher than that of the low- and middle-dose groups (*p* < 0.05), although it was not as high as that of the positive control Ezetimibe group (*p* < 0.05). It is worth noting that the expression of ABCA1 protein and the effect of the ODP-Ia high-dose group were similar to that of the positive control group (no statistical difference).

### 2.7. Effects of GGPP and GW9662 on the Cholesterol Outflow in ODP-Ia-Mediated Foam Cell Cholesterol Efflux

As shown in Figure 7A, the intervention of the LXRα antagonist GGPP significantly inhibited the enhancement of ODP-Ia on cholesterol efflux in foam cells. The previous results indicated that 15 nmol/L ODP-Ia promoted cholesterol efflux in foam cells, resulting in a significant reduction in cholesterol content in ox-LDL-induced foam cells. However, after GGPP intervention, the reduction effect was inhibited, and the cholesterol content in foam cells showed an upward trend. The pre-incubation with a high dose of GGPP (10 mol/L) could efficiently inhibit the effect of ODP-Ia-mediated enhancement, with no significant difference from the foam cell control group, and also no difference with GGPP treatment without ODP-Ia. These results indicate that ODP-Ia-enhancement of cholesterol efflux in foam cells through ODP-1a was suppressed after inhibition of LXRα, which means ODP-Ia needs to activate LXRα nuclear receptors to promote cholesterol efflux in foam cells.

Similarly, as shown in Figure 7B, the PPARγ antagonist GW9662 also significantly interfered with the promotion of ODP-Ia on cholesterol efflux in foam cells. After preincubation with a medium dose of 10 μmol/L GW9662 for 2 h, and then treatment the foam cells with ODP-Ia for 2 h, the enhancement on cholesterol efflux shown previously was significantly reduced. When the dose of GW9662 increased to 20 mol/L, the effect of ODP-Ia on promoting cholesterol efflux was completely inhibited, and there was no significant difference with the foam cell control group. These results suggest that cholesterol efflux function of ODP-Ia in foam cells requires activation of PPARγ nuclear receptors, too.

### 2.8. Effects of GGPP and GW9662 on the Expression of ABCA1, ABCG1, PPARγ and LXRα at mRNA and Protein Level in ODP-Ia-Mediated Foam Cell Cholesterol Efflux

LXRα and PPARγ are very sensitive to macrophage intracellular cholesterol levels. When intracellular cholesterol levels increase, LXRα and PPARγ are activated to regulate downstream-related genes to promote cholesterol excretion. Therefore, in this study, we wanted to clarify the following problems: whether the upregulation of ABCA1 induced by ODP-1a in THP-1 macrophage-derived foam cells is achieved by activating LXRα and PPARγ, and whether the activation is regulated in a cascade manner or activated alone.

As shown in Figure 8, compared with the foam cell control group, the expression of *ABCA1*, *ABCG1*, *PPARγ* and *LXRα* mRNA in Group ODP-Ia increased significantly (*p* < 0.05). However, when GGPP was added, the expressions of those genes were significantly decreased (compared with the ODP-Ia group, *p* < 0.05). Medium-concentration (5 mol/L) GGPP could significantly inhibit the promotion of ODP-Ia on *ABCA1*, *LXRα* and *PPARγ* mRNA expression (compared with the foam cell control group, *p* > 0.05). However, the inhibi-tion of ABCG1 expression only required a low concentration (1 µmol/L). There was no significant dose-dependent effect of GGPP on the expression of these four genes.

In the above results, GGPP, an antagonist of LXRα, reduced the transcription levels of *ABCA1* and *ABCG1* genes, suggesting that LXRα regulates the expression of these two genes in the process of cholesterol efflux. The same regulatory effect was also observed in the experimental group treated with PPARγ antagonist GW9662. As shown in Figure 9, the expression of these four genes decreased significantly in all groups that were pre-incubated with PPARγ antagonist GW9662 (compared with the ODP-Ia group, *p* < 0.05). Low-concentration (5 mol/L) GW9662 could significantly inhibit the promotion of ODP-Ia on the expression of *ABCG1* and *PPARγ* mRNA (compared with the foam cell control group, *p* < 0.05). The inhibition of the expression of *ABCA1* and *LXRα* required a medium concentration (10 µmol/L) and a high concentration (20 µmol/L), respectively. The decreased expression of *LXRα* mRNA indicated that LXRα was also regulated by PPARγ.

As shown in Figure 10, the expression of ABCA1, ABCG1, PPARγ and LXRα protein in Group ODP-Ia was significantly enhanced. However, after pretreatment with different concentrations of GW9662 or GGPP, the electrophoretic bands of those four proteins ex-pression became shallower. The statistical results after quantification of protein expression were basically consistent with the results of the banding analysis. ODP-Ia significantly enhanced the expression of ABCA1, ABCG1, PPAR and LXRα proteins in THP-1 macrophage-derived foam cells (SNK-q test, *p* < 0.05), but the expression of these proteins was significantly reduced after pretreatment with LXR antagonist GGPP and PPARγ antagonist (*p* < 0.05).When the GGPP intervention dose reached a high dose of 10 mol/L, the expression of these proteins all decreased significantly to even lower levels than that of the foam cell group (*p* < 0.05), regardless of whether ODP-Ia pretreatment was used in advance.

These results suggest that ODP-Ia can significantly enhance the expression of ABCA1, ABCG1, PPAR and LXRα proteins in THP-1 macrophage-derived foam cells, but this enhancement can be inhibited or blocked by LXRα antagonist GGPP and PPARγ antagonist GW9662. PPARγ antagonist GW9662 can downregulate the expression of ABCA1, ABCG1 and LXRα at the same time. LXRα antagonist GGPP can downregulate the expression of ABCA1 and ABCG1, suggesting that PPARγ and LXRα regulate the expression of ABCA1 and ABCG1 in a cascade manner.

## 3. Discussion

In this study, we observed that ODP-Ia could significantly upregulate the expression of *PPARγ*, *PPARα*, *LXRα*, *ABCA1*, *ABCG1*, *SR-BI* mRNA and protein, and promoted apoA-I-mediated reduction in cholesterol accumulation in foam cells, although the effect was not as good as that of the positive control, and required a higher dose. It is suggested that ODP-Ia can promote cholesterol efflux by upregulating the expression of these related genes and proteins. The efficiency of cholesterol efflux is an important basis for assessing RCT [28]. It is also believed that the promotion of cholesterol reverse transport by HDL is mainly mediated by promoting cholesterol efflux [29].

At present, domestic and foreign studies have found that a variety of natural active substances have the effect of promoting the outflow of cholesterol from foam cells [30,31,32,33].

There are many factors affecting cholesterol efflux. PPARγ and PPARα are more well-researched receptors that can upregulate the expression of ABCA1 and SR-B1 in foam cells and promote cholesterol efflux [34]. Our results also showed that ODP-Ia up-regulated the expression of these two receptors while mediating cholesterol efflux. Hence, we speculate that these two receptors play a role in the regulation of ABCA1 and ABCG1 expression in the process of ODP-Ia-mediated cholesterol efflux. Our results confirmed the hypothesis that the effect of ODP-Ia upregulating the expression of *ABCA1*, *ABCG1*, *PPARγ* and *LXRα* mRNA and protein was inhibited or blocked by the LXRα antagonist GGPP and the PPARγ antagonist GW9662. In this study, we also observed that PPARγ antagonist GW9662 not only downregulated the expression of *PPARγ* mRNA and protein, but also downregulated the expression of *LXRα* mRNA and protein, confirming that PPARγ plays a role in the PPARγ-LXRα cascade by regulating the downstream *LXRα* gene. The PPARγ-LXRα pathway is an important pathway for the expression of ABCA1 and ABCG1. The study of genistein [35], soy isoflavones [36], Myristica fragrans [37], the dried pericarp of Citrus reticulata Blanco [38], Ginger extract [39] and traditional Chinese medicine Yin-xing-tong-mai decoction [40] confirmed the existence of this pathway. The Tangier disease caused by ABCA1 mutation cannot express ABCA1 protein, and LXRα and PPARγ cannot regulate the reverse transport of cholesterol, which directly proves the regulation of ABCA1 by PPARγ-LXRα pathway [41,42,43].

There are two ways in which PPARγ regulates the expression of downstream genes. First, when PPARγ is activated by ligands, it binds to the specific peroxisome proliferator response element (PPRE) upstream of those genes to regulate their transcription, translation and biological activity. Second, PPARγ can also bind to LXR or RXR to form a dimer; the target gene is regulated by LXR. LXRα is a subtype of LXR receptor, and another subtype is LXRβ. LXRα is closely related to lipid metabolism. Therefore, only LXRα subtype is discussed in this study. The LXRα subtype is highly expressed in macrophages and liver, and was first discovered in the liver cDNA library. At that time, its natural ligand was not found, so it was classified as an orphan receptor. However, subsequent studies have found that LXR has a variety of ligands, including not only endogenous ligands, such as various hydroxylated cholesterols and epoxy cholesterol, but also physiological ligands, such as glucose and sterol. In recent years, a large number of functional substances of plants have been found to activate LXRα, but these substances also activate PPARγ expression at the same time [44]. Therefore, it is difficult to define whether direct activation of LXRα or activation of PPARγ promotes the binding of PPARγ and LXR to dimer.

In the present study, we found that PPARγ antagonist GW9662 not only inhibits PPARγ expression but also inhibits LXRα expression, while also reducing ODP-Ia-mediated intracellular cholesterol efflux and downregulating ABCA1 expression, indicating that PPARγ and LXRα act synergistically to regulate the expression of ABCA1. Studies have shown that LXRα is the target gene of PPARγ [45], especially in macrophage-derived foam cells. PPARγ and LXRα regulate the cholesterol efflux through cascade regulation. Our results also suggest that ODP-Ia cascades upregulate downstream *ABCA1* and *ABCG1* gene expression through the PPARγ-LXRα pathway, thereby promoting cholesterol outflow. It is suggested that ODP-Ia can be further studied as an activator of LXRα and PPARγ to promote its development and utilization. However, we also found that the LXRα antagonist GGPP downregulated the expression of PPARγ. It is suggested that there may be an interaction between PPARγ and LXRα, and the specific mechanism deserves further study.

## 4. Materials and Methods

### 4.1. Chemical Reagents

Fresh tender *Opuntia dillenii* Haw. cladodes of uniform shape and maturity were collected from Donghai Island, Zhanjiang, Guangdong Province, China. ODP-Ia (chromatographically pure, molecular weight [Mr]: 339 kD) was obtained from *Opuntia dillenii* Haw. aqueous extracts by low pressure chromatography, as described previously [46]. Ezetimibe (Dalian Meilun Biotechnology Co., Ltd., Shanghai, China) was taken as a positive control. GW9662 and GGPP were supplied by Merck (Darmstadt, Germany) and Sigma-Aldrich (Vienna, Austria), respectively. Human THP-1 monocytic cells were obtained from Xiangya cell bank (Central South University, Changsha, China). RIPA Lysis Buffer and PVDF membranes were purchased from Hermo Fisher Scientific, (Shanghai, China). Ox-LDL, ApoA-I, Oil red O, Dimethyl sulfoxide (DMSO), phorbol 12-myristate 13-acetate (PMA), 3-(4,5-dimethylthiazol-2-yl)-2,5-diphenyl tetrazolium bromide (MTT), RPMI-1640 medium and other chemicals were purchased from Sigma-Aldrich Co. (Shanghai, China) unless otherwise indicated. The BCA Protein Assay Kit, cholesterol assay kit (E1005), RNeasy kit, DNA reverse transcription kit were purchased from Applygen Technologies Inc. (Beijing, China), Sigma-Aldrich Co. (Shanghai, China), Tiangen Biochemical technology (Beijing, China) and Thermo Fisher Scientific, (Shanghai, China), respectively.

Rabbit anti-PPARγ, Rabbit anti-LXRα, Rabbit anti-ABCG1, Rabbit anti-SR-BI, Mouse anti-ABCA1, Rabbit anti-PPARα, Rabbit anti-GAPDH, HRP-linked anti-rabbit IgG secondary antibody were purchased from Proteintech Group Inc. (Chicago, IL, USA).

### 4.2. THP-1-Derived Macrophages and Foam Cells

THP-1 cells were inoculated in fresh medium (3 mL). Three microliters of PMA (160 µmol/L) were added. After 24 h incubation at 37 °C, most of the cells transformed from suspension to adhesion, indicating the successful induction of macrophages. Subsequently, the cells were incubated with 1640 medium (serum-free) with 0.3% BSA and 50 μg/mL of ox-LDL in a six-well plate at 37 °C for 48 h after the supernatant was replaced. Oil red O staining was used to identify whether the cells were successfully foamed.

### 4.3. Cell Viability Assays

Cell viability was determined using the MTT assay. THP-1 macrophage-derived foam cells were treated with different concentrations of ODP-Ia (0, 5, 10, 20, 40 and 80 nmol/L), as well as ezetimibe (0, 1, 3, 9, and 100 μmol/L). Then, the cells were rinsed with PBS 3 times after the supernatant was removed by centrifugation (800 rpm, 5 min) 24 h later. Then, the cells were incubated with 200 μL of complete medium and 20 μL MTT solution (5 mg/mL of MTT in PBS) in 5% CO_2_ at 37 °C for 4 h. The supernatant was discarded carefully before 150 μL dimethyl sulfoxide (DMSO) was added. After vibration, the absorbance was read at 490 nm using a plate reader ((Multiskan Skyhigh; Thermo Fisher Scientific, Inc., Shanghai, China). Relative cell viability (percent of control) was calculated as follows: (mean OD (490 nm) of treated cells/mean OD (490 nm) of control cells) × 100.

### 4.4. ApoA-I Treatment in ODP-Ia-Induced Foam Cell

After macrophages were induced, the THP-1 macrophage-derived foam cells (10^6^/mL) were treated with different apoA-I concentrations (0, 5, 10, 15, and 20 mg/L) for 24 h, and the total cholesterol content in the cells was measured to investigate the optimal concentration of apoA-I for treatment. Then, this optimal treatment concentration (20 mg/L) was used to treat the foam cells, and the cholesterol efflux of the foam cells was measured at different times (6, 12, 18, 24 and 36 h) to determine the optimal treatment time. Finally, the THP-1 macrophage-derived foam cells (10^6^/mL) were incubated with different ODP-Ia concentrations (0, 5, 10, and 15 nmol/L) for 2 h, and then, apoA-I treatment was continued for 24 h. The lever of cholesterol in the foam cells was measured to investigate the effects of ODP-Ia on cholesterol outflow from apoA-I-treated foam cells during cholesterol efflux.

### 4.5. Groups of the Effect of ODP-Ia on the Cholesterol Efflux and the Associated Gene Expression in Foam Cells

After induction of macrophages, the cells were randomly divided into 6 groups, as shown in Table 1, and each treatment was performed three times, undergoing three replicates each time. The cells in each treatment group were collected, total RNA was extracted and the expression levels of *ABCA1*, *ABCG1*, *SR-BI*, *PPARγ*, *PPARα* and *LXRα* mRNA were detected by RT-qPCR. Western blot was used to detect their protein expression.

### 4.6. PPARγ-LXRα Antagonist Treatment in ODP-Ia-Induced Foam Cell

Intervention of THP-1 -derived foam cells with 15 nmol ODP-Ia was performed with different concentrations of LXRα antagonist GGPP and PPARγ antagonist GW9662, respectively. The total cholesterol content in the cells was measured to investigate whether GGPP and GW9662 inhibited the promotion of ODP-Ia on total cholesterol efflux in foam cells. In order to examine whether GGPP and GW9662 could inhibit the upregulation of *ABCA1*, *ABCG1*, *PPARγ* and *LXRα* expression level induced by ODP-1a in the foam cells, the mRNAs were detected by RT-qPCR. Western blot was used to detect those protein expressions. The experimental design of the LXRα antagonist GGPP were shown in Table 2. The PPARγ antagonist GW9662 was designed in a similar way, but the concentrations of GW9662 were 5, 10 and 20 mol/L, respectively.

### 4.7. Determination of Cholesterol Content in THP-1 Macrophage-Derived Foam Cells

Total cholesterol content of the foam cells was measured through an enzymatic assay technique using a commercial cholesterol assay kit. All experiments were performed according to the manufacturer’s recommended protocol.

### 4.8. Western Blot Analysis

After the cells were lysed with RIPA Lysis Buffer, and total protein concentrations in the cells were examined using BCA Protein Assay Kits. The total protein content of each sample was 10 μL. The sample was mixed with the sample buffer in proportion and boiled for 5 min. Then, 20 μL of the mixture was used for SDS-PAGE electrophoresis (the electrophoretic voltage was 80 V for the shrink gel and 120 V for the separation gel). Proteins separated by the separating gel were transferred onto PVDF membranes. Then, the membranes were blocked with free-fat milk in TBS at 4 °C for 1 h and incubated with primary antibodies for PPARγ (1:1000 dilution), PPARα (1:1000 dilution), LXRα (1:1000 dilution), ABCA1 (1:200 dilution), ABCG1 (1:1000 dilution), SR-B1(1:1000 dilution), and GAPDH (1:1000 dilution) at 4 °C overnight. At last, the membranes were incubated with HRP-conjugated antibodies for 1 h at room temperature. After incubation, the membranes were washed 3 times with TBST for 15 min each time. The ECL color exposure test was performed. The exposed film was scanned and analyzed using quantity one professional grayscale analysis software. Using GAPDH as the internal reference, histone expression of each treatment was calculated and compared.

### 4.9. RT-PCR Assays

Total RNA was extracted from macrophages using an RNeasy kit, and cDNA was prepared using a cDNA reverse transcription kit. Then, 10 μL of the reverse transcription product was used for the PCR cycle. Primer and probe designs are presented in Appendix A. The reaction conditions were pre-denaturation at 95 °C for 10 min, denaturation at 95 °C for 15 s, and annealing extension at 60 °C for 60 s. This cycle was repeated 40 times. According to the experimental results, the amplification efficiencies of the target gene and the internal reference gene GAPDH were basically the same. Therefore, the experimental data could be analyzed using the 2^−ΔΔCt^ method [47] to calculate the expression levels of each gene relative to GAPDH.

### 4.10. Statistical Analysis

All experiments were repeated at least three times, and the values are expressed as the mean ± SD. The results were analyzed using one-way analysis of variance (ANOVA) with SNK-q test, and a *p*-value < 0.05 was considered statistically significant.

## 5. Conclusions

In conclusion, our data reveal a favorable role of ODP-Ia in promotion of cholesterol efflux in the foam cells. Mechanistically, ODP-Ia activates the PPARγ/LXRα pathway to enhance ABCA1- and ABCG1-dependent cholesterol efflux. These findings extend our understanding for the anti-atherogenic action of ODP-Ia, and further support the notion that ODP-Ia may be a promising drug candidate for therapeutic intervention of atherosclerotic cardiovascular disease.

## Figures and Tables

**Figure 1 molecules-27-08639-f001:**
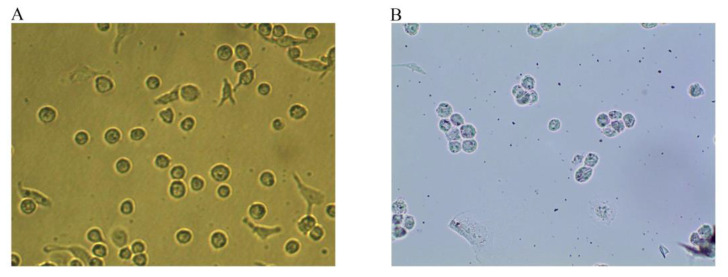
Macrophages and Oil red O staining maps of foam cells (Magnification, ×100). (**A**) After 24 h of PMA stimulation, the cells differentiated into macrophages. (**B**) After 48 h of incubation with 50 ng/mL ox-LDL, the macrophages developed into foam cells, and intracellular lipid droplets were stained with Oil red O.

**Figure 2 molecules-27-08639-f002:**
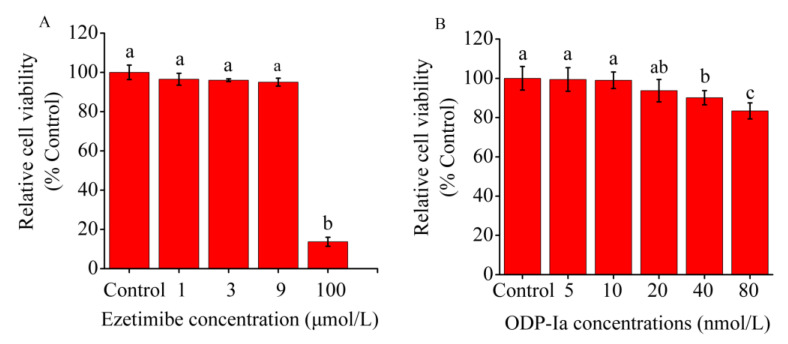
Effect of ODP-Ia and ezetimibe on THP-1 macrophage-derived foam cell viability, as determined through MTT analysis. Values are mean ± SD, *n* = 5. (**A**) ODP-Ia. (**B**) Ezetimibe. Different superscript letters indicate multiple comparisons with significant differences (SNK-q test, *p* < 0.05).

**Figure 3 molecules-27-08639-f003:**
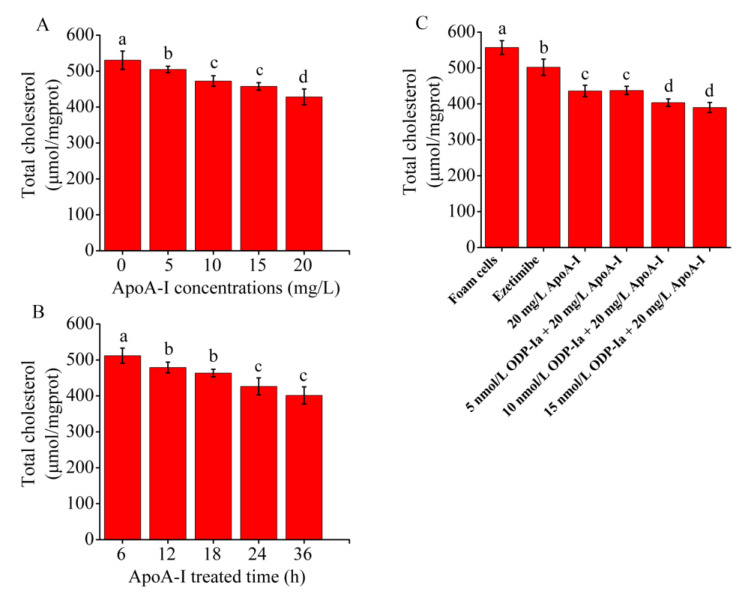
Effects of apoA-I on cholesterol accumulation in THP-1 macrophage-derived foam cells. Values are mean ± SD, *n* = 5. (**A**,**B**) Total cholesterol content of THP-1 macrophages decreased in a time- and concentration-dependent manner in the apoA-I-treated foam cells. (**C**) apoA-I-induced cholesterol efflux mediated by ODP-Ia. Different superscript letters indicate multiple comparisons with significant differences (SNK-q test, *p* < 0.05).

**Figure 4 molecules-27-08639-f004:**
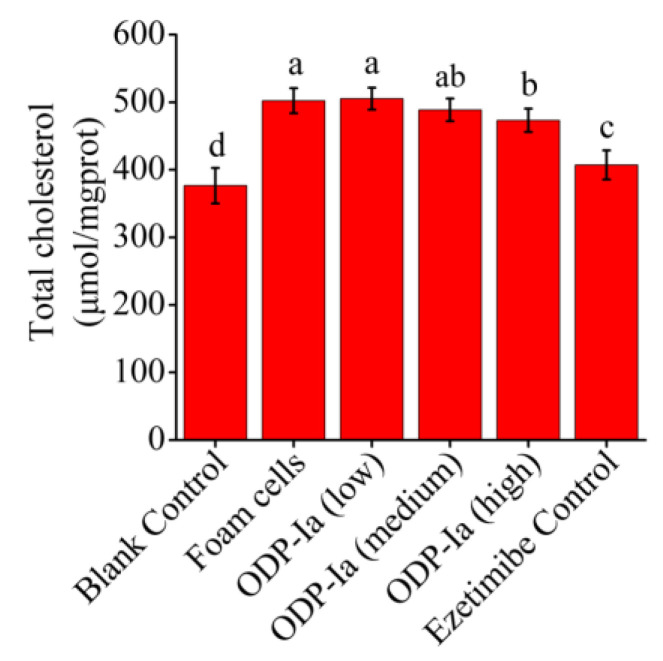
Effects of ODP-Ia on the cholesterol outflow in foam cells. (mean ± SD, *n* = 5). Different superscript letters indicate multiple comparisons with significant differences (SNK-q test, *p* < 0.05).

**Figure 5 molecules-27-08639-f005:**
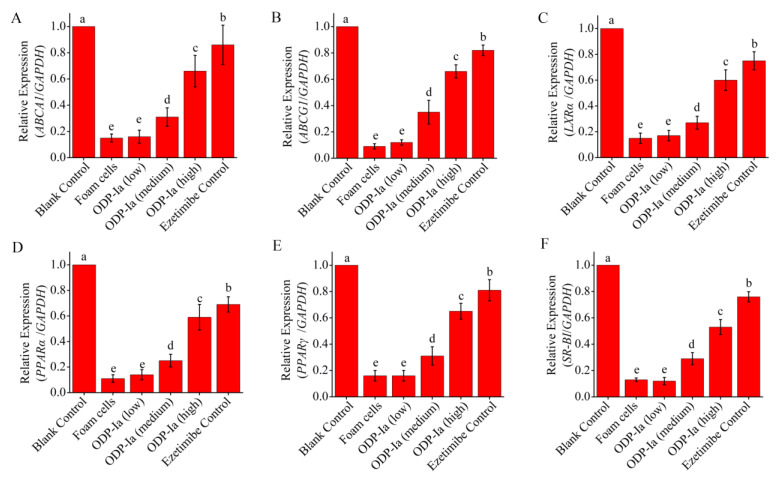
mRNA expression of cholesterol efflux-associated gene mediated by ODP-Ia. Values are mean ± SD, *n* = 9. (**A**) *ABCA1* mRNA. (**B**) *ABCG1* mRNA. (**C**) *LXRα* mRNA. (**D**) *PPARα* mRNA. (**E**) *PPARγ* mRNA. (**F**) *SR-BI* mRNA. Different superscript letters indicate multiple comparisons with significant differences (SNK-q test, *p* < 0.05).

**Figure 6 molecules-27-08639-f006:**
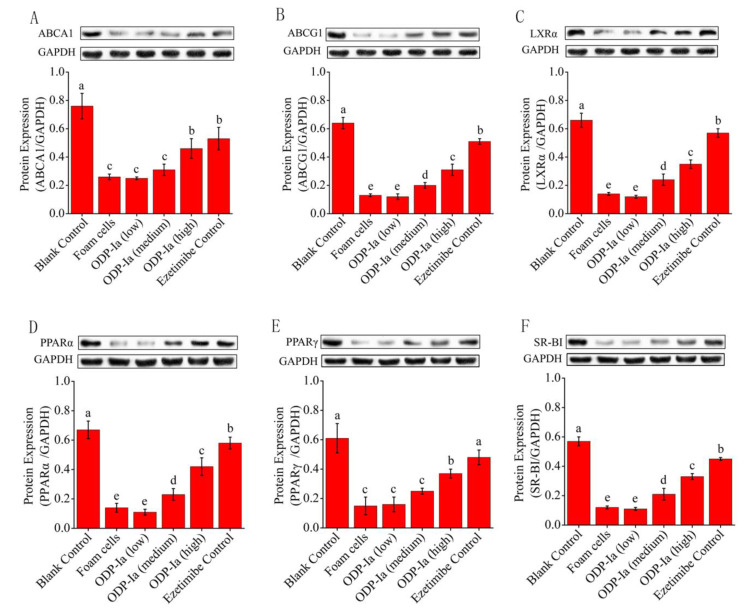
Effect of ODP-Ia on protein expression of ABCA1, ABCG1, LXRα, PPARα, PPARγ and SR-BI in THP-1 macrophage-derived foam cells. Dates in figure represent the density values of each band (mean ± SD, *n* = 9). (**A**) ABCA1 protein expression level. (**B**) ABCG1 protein expression level. (**C**) LXRα protein expression level. (**D**) PPARα protein expression level. (**E**) PPARγ protein expression level. (**F**) SR-BI protein expression level. Different superscript letters indicate multiple comparisons with significant differences (SNK-q test, *p* < 0.05).

**Figure 7 molecules-27-08639-f007:**
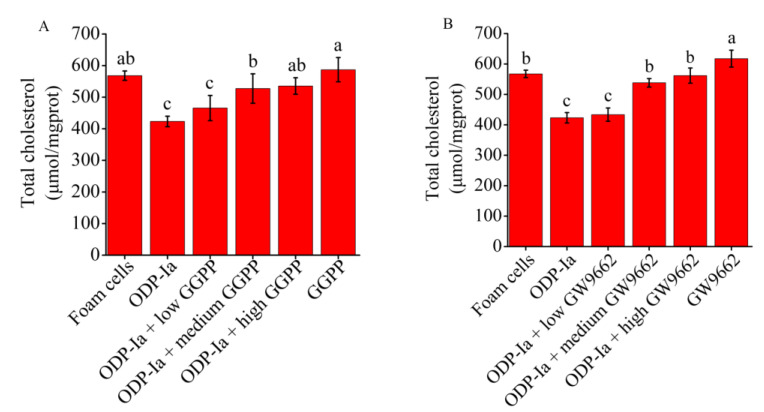
The ODP-Ia-induced cholesterol efflux mediated by GGPP (**A**) and GW9662 (**B**) (mean ± SD, *n* = 5). Different superscript letters indicate multiple comparisons with significant differences (SNK-q test, *p* < 0.05).

**Figure 8 molecules-27-08639-f008:**
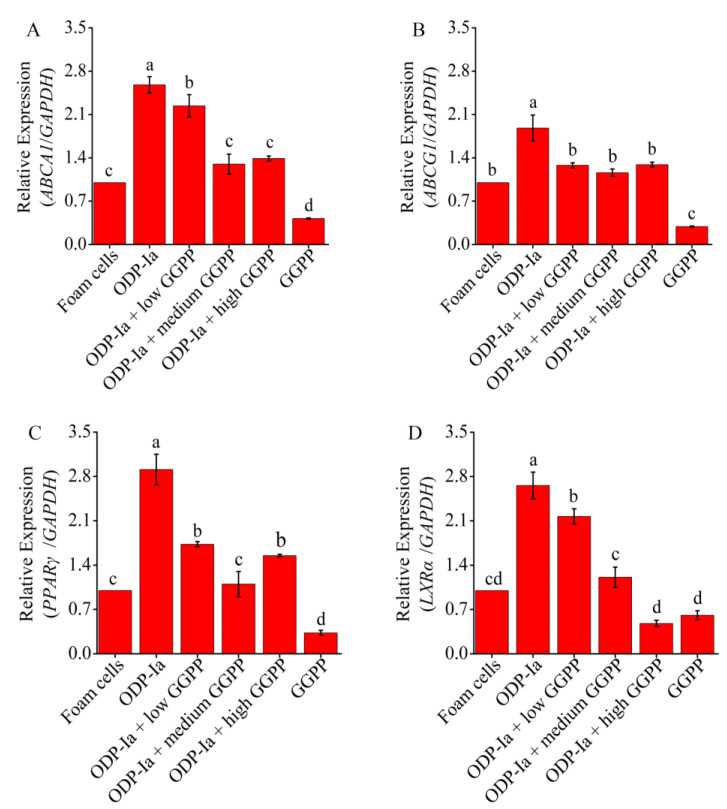
mRNA expression of cholesterol efflux associated gene mediated by GGPP and ODP-Ia. (mean ± SD, *n* = 3). (**A**) *ABCA1* mRNA. (**B**) *ABCG1* mRNA. (**C**) *LXRα* mRNA. (**D**) *PPARγ* mRNA. Different superscript letters indicate multiple comparisons with significant differences (SNK-q test, *p* < 0.05).

**Figure 9 molecules-27-08639-f009:**
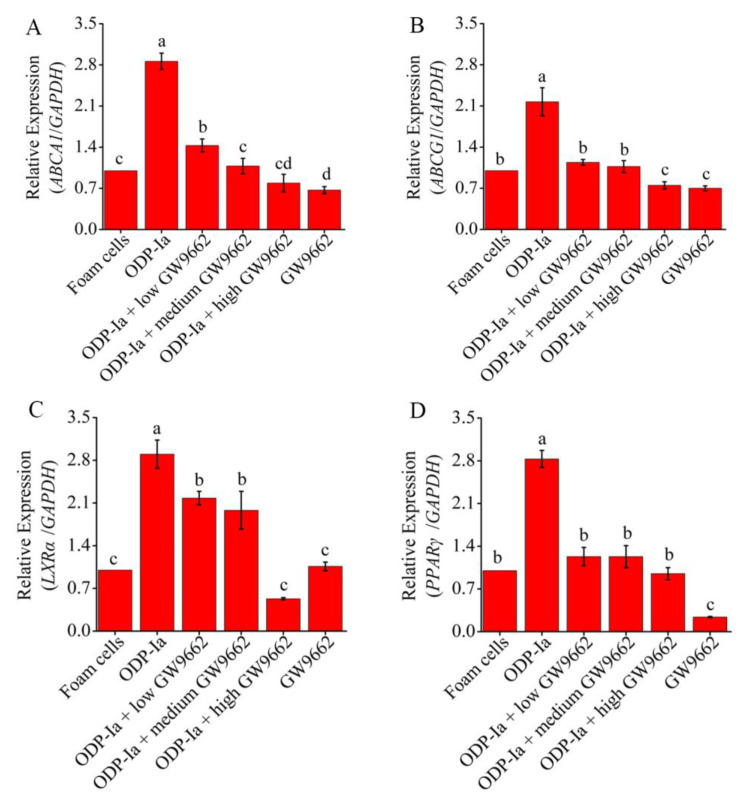
mRNA expression of cholesterol efflux associated gene mediated by GW9662 and ODP-Ia. (mean ± SD, *n* = 3). (**A**) *ABCA1* mRNA. (**B**) *ABCG1* mRNA. (**C**) *LXRα* mRNA. (**D**) *PPARγ* mRNA. Different superscript letters indicate multiple comparisons with significant differences (SNK-q test, *p* < 0.05).

**Figure 10 molecules-27-08639-f010:**
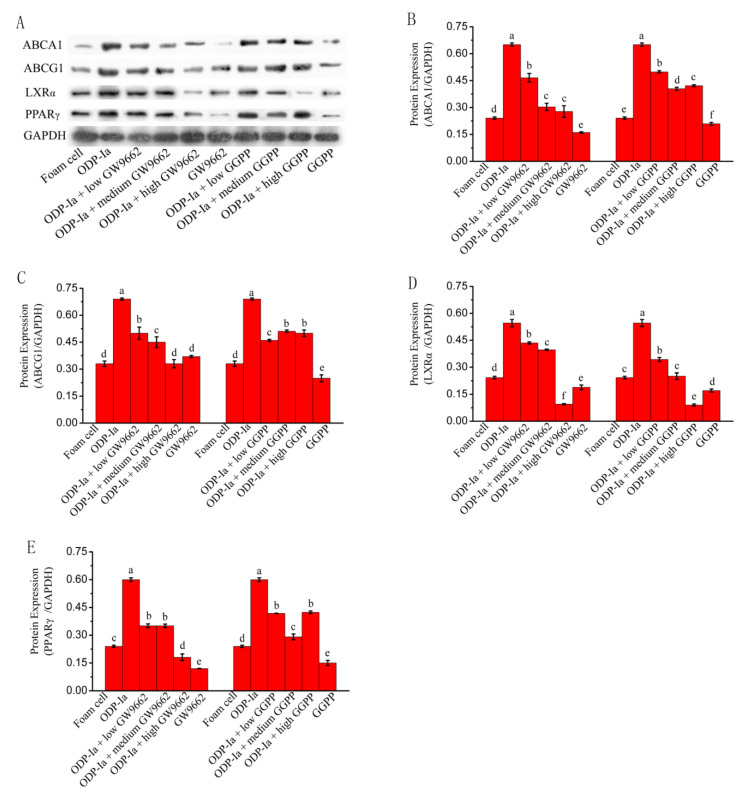
Protein expression of cholesterol efflux-associated gene mediated by GW9662 and GGPP. Dates in figure represent the density values of each band (mean ± SD, *n* = 9) (**A**) Western blot analyses of ABCA1, ABCG1, LXRα and PPARγ in THP-1-derived foam cells. (**B**) ABCA1 protein expression level. (**C**) ABCG1 protein expression level. (**D**) LXRα protein expression level. (**E**) PPARγ protein expression level. Different superscript letters indicate multiple comparisons with significant differences (SNK-q test, *p* < 0.05).

**Table 1 molecules-27-08639-t001:** Groups of the effect of ODP-Ia on the cholesterol efflux-associated gene expression in foam cells.

Group	Treatment
Blank	no ox-LDL is added.
Foam cells	50 ug/mL ox-LDL
ODP-Ia (low dose)	5 nmol/L ODP-Ia + 50 µg/mL ox-LDL
ODP-Ia (medium dose)	10 nmol/L ODP-Ia + 50 µg/mL ox-LDL
ODP-Ia (high dose)	15 nmol/L ODP-Ia + 50 µg/mL ox-LDL
Ezetimibe positive control	3 mmol/L ezetimibe + 50 µg/mL ox-LDL

**Table 2 molecules-27-08639-t002:** Scheme of the study on the effect of GGPP on cholesterol efflux and its associated gene expression in foam cells treated with ODP-Ia.

Group	GGPP (µmol/L)	ODP-Ia (15 mmol/L, 2 h)	ox-LDL (50 µg/mL, 48 h)	apoA-I (20 mg/L, 12 h)
Foam cell group	-	-	+	+
ODP-Ia	-	+	+	+
ODP-Ia with low GGPP	1	+	+	+
ODP-Ia with medium GGPP	5	+	+	+
ODP-Ia with high GGPP	10	+	+	+
GGPP group	10	-	+	+

## Data Availability

The data presented in this study are available on request from the corresponding author.

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
