# Peer review of "Opuntia dillenii Haw. Polysaccharide Promotes Cholesterol Efflux in THP-1-Derived Foam Cells via the PPARγ-LXRα Signaling Pathway"

_molecules, 2022, doi:10.3390/molecules27248639_

Round 1

Reviewer 1 Report

Thank you for submitting the manuscript “ODP-Ia promotes cholesterol efflux in THP-1 derived foam cells via PPARγ-LXRα signaling pathway” to Molecules. The authors studied the effect of Opuntia dillenii Haw. Polysaccharide in cholesterol efflux and it really looks like the results are positive. However, the study is still at a cellular scale and needs to be replicated in animals and I suggest that this weakness be added to the text. Also, I have small considerations:

Title: please define ODP-Ia. Avoid abbreviations in the title.

Line#21: added and after “ABCA1”

Line#23: in fact, more in vivo trials are needed to reach this conclusion.

Line#25: comma or semicolon?

Line#41: define the abbreviations. Apply throughout the manuscript.

Lines#65-68: Please rewrite this part of the text.

Line#365: correct all scientific names to italics

Line#408: add comma before respectively.

Line#423: add space after Table

Line#429: Was the low, medium and high dose established based on previous work? Please explain further.

Reviewer 2 Report

In this article, the authors observed that Opuntia dillenii Haw. polysaccharide (ODP-Ia) could promote cholesterol efflux via PPARγ-LXRα signaling pathway in THP-1 foam cells. However, many unclear issues need to be addressed and my detailed comments are as follows:

1.     In “4. Materials and Methods”, the methods of “ Measurement of total cholesterol content”, “RT -qPCR”, “ Western blot”, and “Statistic” are not described in detail. It is recommended to describe the detailed methods. In addition, many materials are not labeled as sources, such as ox-LDL, 1640 medium, apoA-I, etc. Please check and complete the materials carefully.

2.     Missing from the experimental design was a test to identify whether THP-1 had successfully differentiated into macrophages. In addition, Oil red O Staining was mentioned in the experiment, but staining results of foam cells were not given in the experimental results, which could not determine whether foam cells were formed or not.

3.     In part “4.3”, it is desirable to add an experimental design to examine the effect of ODP-Ia on the cholesterol efflux at different time points after ox-LDL stimulation of macrophages.

4.     In part “4.4”, It is recommended to add experimental groups without apoA-I to verify the effect of apoA-I on cholesterol efflux.

5.     In lines 136-137, The sentence after line 136 is not visible. Please check if there are any mistakes.

6.     In lines 396-389, the authors found that the LXRα antagonist GGPP also down-regulated the expression of PPARγ, and there was a dose relationship. However, in the experimental results, mRNA expression and protein levels of THP-1 foam cells treated with ODP-Ia and high GGPP were significantly higher than those treated with ODP-Ia and medium GGPP, which was contrary to the finding of “there was a dose relationship ”. This is a contradiction between them, and the authors should explain it.

7.     In parts “2.4” and “2.5”, the interpretation of the results is too miscellaneous and is suggested to be simplified.

Round 2

Reviewer 2 Report

Thank you for answering my comments and questions. I think there are no more questions now.